# Prospective Study of 4 Gy Radiotherapy for Orbital Mucosa-Associated Lymphoid Tissue Lymphoma (FORMAL)

**DOI:** 10.3390/cancers14174274

**Published:** 2022-09-01

**Authors:** Jaehyeon Park, Ji Woon Yea, Se An Oh, Min Kyoung Kim, Jun Hyuk Son, Jae Won Park

**Affiliations:** 1Department of Radiation Oncology, Yeungnam University College of Medicine, 170, Hyeonchung-ro, Nam-gu, Daegu 42415, Korea; 2Division of Hematology-Oncology, Department of Internal Medicine, Yeungnam University College of Medicine, Daegu 42415, Korea; 3Department of Ophthalmology, Yeungnam University College of Medicine, Daegu 42415, Korea

**Keywords:** MALT lymphoma, radiotherapy, conjunctiva, ocular adnexa, orbit, non-Hodgkin lymphoma

## Abstract

**Simple Summary:**

Marginal zone B-cell lymphoma of mucosa-associated lymphoid tissue (MALToma) is a slow-growing lymphoma with a good prognosis. This study was designed to evaluate the effectiveness of radiotherapy at a very low dose of 4 Gy (2 Gy × 2 fractions) in stage I orbital MALToma. Patients with complete remission after a very low dose of 4 Gy (2 Gy × 2 fractions) radiotherapy were closely monitored, and those who did not achieve remission received an additional 24 Gy radiotherapy. Using 4 Gy radiotherapy for orbital MALToma, 11 out of 17 lesions achieved complete remission. There was no transformation of diffuse large B-cell lymphoma, and there was only one local failure. Radiation therapy at a low dose of 4 Gy could be performed effectively and safely with a planned second-line treatment.

**Abstract:**

External beam radiotherapy is effective for stage I orbital mucosa-associated lymphoid tissue lymphoma (MALToma). Hence, very-low-dose radiotherapy is increasingly being investigated. We conducted a single-center prospective phase II trial to evaluate the effectiveness of very-low-dose radiotherapy of 4 Gy (2 Gy × 2 fractions) in pathologically confirmed stage I orbital MALToma. In this first prospective trial, patients with complete response were observed after 3–6 months of follow-up. For patients without complete remission, a radiation dose of 24 Gy/12 fractions was additionally delivered. The primary endpoint was complete response rate; secondary endpoints were overall survival, local control, and progression-free survival. Seventeen patients were screened and three patients refused enrollment during October 2018–October 2021. Thus, 14 patients (17 eyes) were analyzed (median follow-up, 28.2 months). The overall response rate was 100% (complete remission: 11 lesions; partial remission: six lesions). In all lesions with residual disease, additional radiation therapy (dose: 24 Gy) was performed. One local failure was observed. Therefore, 4 Gy ultralow-dose radiation therapy for orbital MALToma was safely performed with a planned second-line treatment in patients without complete remission. This is the first prospective study to report the effectiveness of ultralow-dose radiotherapy of 4 Gy for stage I orbital MALToma treatment.

## 1. Introduction

Marginal zone B-cell lymphoma of mucosa-associated lymphoid tissue (MALToma) accounts for approximately 7–8% of all non-Hodgkin’s lymphomas (NHLs) [1]. MALToma is an indolent tumor and has a very favorable prognosis [2]. The most frequently involved non-gastric sites are the conjunctiva and ocular adnexa, and most cases of the disease are stage I [3]. External beam radiotherapy is a highly effective treatment for stage I orbital MALToma, and the local control rate of the treatment is reported to be approximately 80–90% [4,5,6]. On the basis of the results of a phase III study, a radiation dose of 24 Gy is considered sufficient as a therapeutic dose [7].

The outcomes of standard dose radiation therapy indicate a sufficient local control rate. Accordingly, lower doses of this therapy are gaining interest in the clinical setting. A comparison of 4 Gy (2 Gy × 2 fractions) and 24 Gy (2 Gy × 12 fractions) doses for indolent lymphoma was reported in the FORT trial [8,9]. Radiation therapy of a 4 Gy dose showed a lower local control rate than that of a 24 Gy dose, but there was no difference in the overall survival (OS) rate. Furthermore, the complete remission rate in the 4 Gy arm of the trial was nearly 55%. However, most of the cases assessed in this study pertained to follicular lymphoma (86%). Considering that the proportion of stage I patients was 42% in this study, the number of stage I orbital MALToma patients was considered to be very small.

Since the lens of the eye is the most sensitive to radiation, it is more important to lower the radiation dose in the orbit. A retrospective study showed excellent local control of indolent orbital lymphomas at a dose of 4 Gy [10]. The complete remission rate was 85%, and the 2 year local progression-free survival (PFS) rate for patients who achieved complete remission was 96%. However, this study was retrospective and included various histologic types, with clinical stages ranging from stage I to IV. 

Therefore, in our institution, we conducted a prospective study using 4 Gy low-dose radiation therapy, followed by close observation through follow-ups in cases of complete remission. Furthermore, we administered an additional 24 Gy of radiation therapy in cases without complete remission and reported the initial results.

## 2. Materials and Methods

### 2.1. Study Design 

This was a single-center prospective trial designed to evaluate the effectiveness of radiotherapy at a very low dose of 4 Gy (2 Gy × 2 fractions) in stage I orbital MALToma. The study was initiated from October 2018. The study protocol is shown in Figure 1. After 3–6 months of follow-up, patients with complete response were closely observed. For patients with partial response or stable disease/progressive disease, radiation therapy of 24 Gy/12 fractions was additionally delivered.

Patients were eligible for assessment in this study if they had a histologically confirmed diagnosis of MALTomas localized in the orbit (Ann Arbor stage I) and could receive radiotherapy with curative intent. The patients were ≥18 years of age. Patients who did not cooperate with radiation therapy, had uncontrolled serious underlying disease, were diagnosed with another cancer within 5 years, or had a history of other radiation therapy in the orbit were excluded from the study.

This study was approved by the Institutional Review Board of Yeungnam University Medical Center (YUMC 2018-03-003). All the patients provided informed consent to participate in this study.

### 2.2. Staging Workup and Radiotherapy Technique

Staging workup included physical examination, contrast-enhanced computed tomography (CT) (head and neck, chest, abdomen, and pelvis), and 18-fluorodeoxyglucose-positron emission tomography-CT (FDG-PET-CT). Physical examination was performed by an experienced ophthalmologist. Magnetic resonance imaging (MRI) and bone marrow biopsy were also performed as needed. 

Gross tumor volume (GTV) was measured from the visible mass in an imaging study (CT, MRI, or PET-CT). If GTV was not visible in the imaging study, it was not delineated. Clinical target volume (CTV) for the conjunctiva was the whole conjunctiva, and that for the retrobulbar region was the whole retrobulbar tissue. When the mass invaded both the conjunctiva and the retrobulbar areas, the whole globe was treated. A thermoplastic mask was used for immobilization. In conjunctival MALToma, a 6 or 9 MeV electron beam was used with Superflab bolus and commercial lens shielding. The electron block was manufactured such that it had a margin of 1.5 cm from the CTV. In a tumor with a retrobulbar mass with or without the conjunctiva, planning target volume was generated with a margin 0.5 cm from the CTV. In CTV with the retrobulbar area only, intensity-modulated radiotherapy was delivered for lens protection. In CTV for the whole globe, three-dimensional radiotherapy with a 6 MV photon beam was delivered (Figure 2). Eclipse 15.6 (Varian, Palo Alto, CA, USA) was used for treatment planning.

### 2.3. Assessment of Response and Follow-Up

Response evaluation was performed at 1, 3, and 6 months after the end of the treatment. If there were residual lesions for up to 6 months, then radiotherapy was performed according to the protocol, and a follow-up schedule was planned for response evaluation again for 6 months. After response evaluation, regular follow-up was conducted every 3 months until 2 years after the end of treatment; thereafter, it was carried out every 6 months. In the case of conjunctival MALToma with no visible mass in the imaging study, response evaluation and follow-ups were performed through examination by an experienced ophthalmologist. In cases in which a mass was observed, it was judged by the ophthalmologist by combining the examination and CT findings. The clinical response was evaluated according to the RECIST 1.1 (Response Evaluation Criteria in Solid Tumors) [11]. Toxicity was assessed on the basis of NCI-CTCAE 4.0.3.

### 2.4. Statistical Analysis

This study was designed as a phase II trial to evaluate radiation therapy of 4 Gy dose for stage I orbital MALToma. Assuming that the complete remission rate in previous studies was 55% [8,9], a radiation dose of 24 Gy with curative intent could be avoided when a complete remission rate of 40–70% was obtained in this study. Accordingly, the results of the present study were achieved. In this case, a total of 28 patients were required for the statistical power to be 90% at α = 0.10.

The primary endpoint was complete response rate at 6 months after 4 Gy radiotherapy, and secondary endpoints were OS, local control, and PFS. The toxicity profile was also evaluated. Kaplan–Meier survival curves were constructed and used for calculating OS and PFS.

Patient enrollment was carried out for 4 years. Considering a dropout rate of 10%, we planned to include 32 people, but only 15 people enrolled by the planned end date owing to a slow accrual. Therefore, we conducted a review of the treatment results at this point to determine if the study period was to be extended.

## 3. Results

### 3.1. Patient Characteristics 

A total of 17 patients were screened from October 2018 to October 2021, and three patients refused to enroll in this study. Therefore, a total of 14 patients were analyzed (Table 1). Because three patients had lesions in both eyes, 17 eyes were enrolled. Nine eyes were on the left side, and 14 eye lesions involved the conjunctiva only (Table 2). Four lesions were visible on contrast-enhanced CT, and seven lesions were hypermetabolic lesions, as observed with the FDG-PET-CT examination.

### 3.2. Response and Clinical Outcomes

The median follow-up duration was 28.2 months. The complete remission was achieved in 11 (64.7%) lesions, and partial remission was achieved in six lesions (35.3%). The overall response rate was 100%. For all the lesions with residual disease, additional radiation therapy at a dose of 24 Gy was performed as planned. Complete remission was observed in all patients who received the additional radiotherapy. 

The 2 year OS rate was 100%, and the 2 year PFS rate was 90% (Figure 3). The 2 year local control rate for 17 eyes was 90.6% (Figure 4). Only one local failure was observed. After recurrence, salvage radiotherapy (24 Gy/12 fractions) was performed, and complete remission was achieved again. The patient had no evidence of disease at the last follow-up. 

Grade 1 eyelid swelling was seen in five patients. Dry eye was of grade 1 in one patient and grade 2 in three patients. All cases of grade 1 eyelid swelling and grade 2 dry eye occurred in patients who received additional radiation therapy at a dose of 24 Gy. There were no cases of radiation-induced cataracts.

## 4. Discussion

The present study showed that 4 Gy ultralow radiation therapy for orbital MALToma could be safely performed and results in complete remission in most patients, reserving a second-planned higher dose only in cases with residual lesions.

Among orbital lymphomas, MALToma is the most common pathological type [12]. Orbital MALToma has a slow disease progression and is known to respond well to radiotherapy. In previous studies on radiotherapy, the complete remission rate and local control rate were nearly 95–100%, and distant metastasis was reported to occur very rarely. The role of definitive radiation therapy in MALToma was reported in 90–100% of previous studies [6,13,14,15,16,17,18]. Additionally, in the case of relapse in these studies, complete remission and local control were achieved again through re-irradiation. The radiation treatment dose varied from 24 to 36 Gy in most cases, and the studies that used a dose of more than 24 Gy were successful.

As the lens is the weakest and most sensitive organ to radiation, a dose range of 4–10 Gy can induce cataracts even when the irradiation is divided across 3 weeks to 3 months, which is the general period for radiation treatment [19]. Although the incidence of cataracts differs depending on whether or not the lens was shielded and the type of irradiation method used, Hata et al. [15] and Goda et al. [13] reported that cataract incidence rates were 15–30% even after shielding. Moreover, dry eye was reported in 17–33% of cases when doses of 24–30 Gy were applied, which is a dose regimen that is currently widely used [17,20]. Furthermore, when the tumor invades the conjunctiva and soft tissues behind the eyeball, it is difficult to protect the lens. Therefore, it is considered important to try to lower the radiation dose to the orbital lymphoma.

The 4 Gy (2 Gy × 2 fractions) radiotherapy showed an acceptable clinical response in previous clinical studies [21,22,23]. On the basis of this clinical evidence, the FORT study compared 4 Gy and 24 Gy radiation therapy doses for indolent lymphomas [8,9]. Although there was no difference in OS with the 4 Gy dose, the authors recommended a dose of 24 Gy as the standard dose, since the PFS was lower. However, in this study, most of the tumors were follicular lymphomas, and the number of stage I orbital MALToma was very small. König et al. [24] pointed out that the FORT study may be difficult to apply to small-sized orbital indolent lymphoma because the size or risk classification of the 2 × 2 Gy treatment group is not appropriate.

König et al. [25] compared 52 patients who received conventional radiotherapy from 24 to 46 Gy with seven patients who received low-dose radiotherapy of 4 Gy. The 2 year local progression-free survival of the patients who underwent low-dose radiotherapy was 100%, and the conventional radiotherapy group was 93.5%, showing good clinical outcomes in both groups. Carolina et al. reported a 4 Gy radiotherapy dose for indolent orbital lymphomas retrospectively [10]. Of the 20 patients (27 eyes), 11 had follicular lymphoma and eight had MALToma. There were seven stage I patients. The overall response rate was 96%. There were no further recurrences in 23 patients with complete remission.

At Memorial Sloan Kettering Cancer Center, a retrospective study reported about 4 Gy radiation therapy for indolent lymphomas [26]. Of the total 299 patients, 52 were potentially curable, and the local progression rate in these patients was 9%, with lower PFS and complete remission rates compared to the patients who received the 24 Gy dose. However, most of these patients had follicular lymphomas and few were orbital MALTomas. The local recurrence rate in patients who had complete remission within 6 months was 2.9%. The diffuse large B-cell lymphoma transformation rate was 9% in all patients but zero in potentially curable patients. After treatment, 12 patients with residual lesions within 8–12 weeks underwent additional radiotherapy. Among these patients, two had local progression and seven had distant progression.

To the best of our knowledge, this is the first prospective study to verify the effectiveness of ultralow-dose radiotherapy of 4 Gy in stage I orbital MALToma with a curative intent. Radiation therapy of a dose of 4 Gy is very convenient and can minimize toxicity. At this radiation dose, radiation-induced cataract might be minimal or might not occur when the whole globe is irradiated. In the present study, complete remission was achieved within 6 months at a dose of 4 Gy in 11 of the 17 eyes that were treated, and no additional radiation therapy was provided to these patients. A complete remission rate of 64.7% was obtained, which was higher than the expected results at the beginning of the study. All the patients without complete remission received an additional planned radiotherapy. Only one patient experienced local failure and was successfully treated with re-irradiation.

In this study, when only a dose of 4 Gy was irradiated, no special side-effects were observed during or after treatment because the dose was very low. No serious adverse events of grade 3 or higher were observed. In addition, grade 1 or 2 toxicities were also identified only in patients with residual lesions who received 24 Gy of additional radiation therapy. Kim et al. reported that 24–36 Gy radiation therapy for orbital MALToma might cause dry eyes due to effects on meibomian glands [27]. In addition, no cataracts were observed. In all patients with conjunctival MALToma, lens shielding was used when electron beam therapy was used, and, in the case of retrobulbar tumor, the lens was sufficiently protected with IMRT. However, there were cases where both retrobulbar and conjunctiva were involved in two patients. One of these patients achieved complete remission at 4 Gy, and no cataract occurred until 40 months follow-up. Another patient did not develop a cataract until 8 months follow-up after receiving an additional 24 Gy of radiation, which was likely due to the short follow-up effect.

This study had several limitations. First, because of a slow accrual, the expected number of enrolled patients was not met during the study period. Although 28 patients were required to have a statistical power of 90%, a total of 14 patients (17 lesions) were enrolled. Therefore, when the expected results were obtained, the initial data were analyzed to determine whether additional patients should be enrolled to obtain the required statistical power. The response rate and retreatment rate expected from the initial data were recorded, and since only one recurrence had been observed thus far, the current protocol was considered safe. Moreover, unlike previous studies, the present study was conducted on pathologically confirmed orbital MALToma patients only; therefore, the findings may have clinical implications despite the relatively small number of patients. Second, because complete remission was not pathologically confirmed during the response evaluation, there was a possibility that the retreatment rate would be relatively high. Even if sufficient partial remission was observed, additional radiation therapy was performed as planned if residual lesions were clinically suspected during the treatment. There was a possibility that patients who responded well could have an excellent clinical result without the need for additional treatment. Third, the evaluation of late toxicity was limited because the follow-up of the study was short. Therefore, we plan to conduct additional studies by collecting more patients in the future and extending the follow-up period. In addition, since the toxicity was evaluated with NCI-CTCAE grade, not with an ophthalmic special test, there may be limitations in the interpretation of the results.

## 5. Conclusions

In conclusion, this novel prospective trial showed that 4 Gy ultralow-dose radiation therapy for orbital MALToma could be safely performed when a planned second-line treatment was performed in patients who had not achieved complete remission. Furthermore, 64.7% of the eyes had local control, without the need for additional radiotherapy (only 4 Gy). Since the number of patients enrolled was too small to obtain the required statistical power, and since there may be limitations in the evaluation of late toxicity with a short follow-up, we decided to conduct a further investigation by recruiting patients for an additional study period.

## Figures and Tables

**Figure 1 cancers-14-04274-f001:**
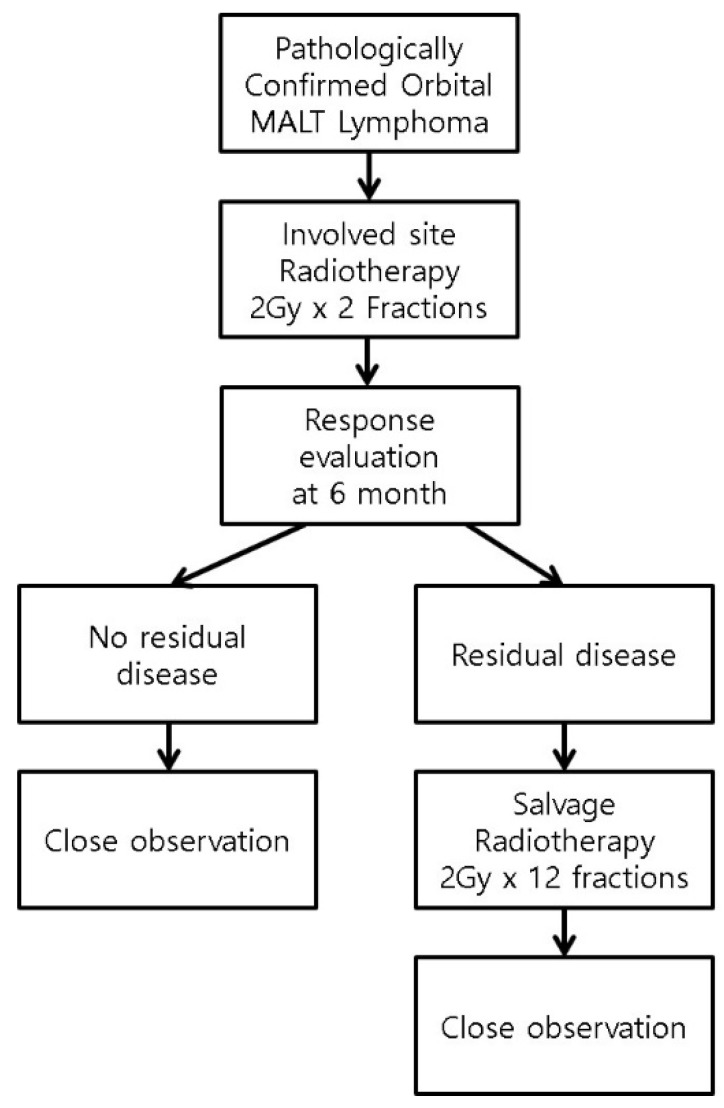
Image showing the treatment protocol.

**Figure 2 cancers-14-04274-f002:**
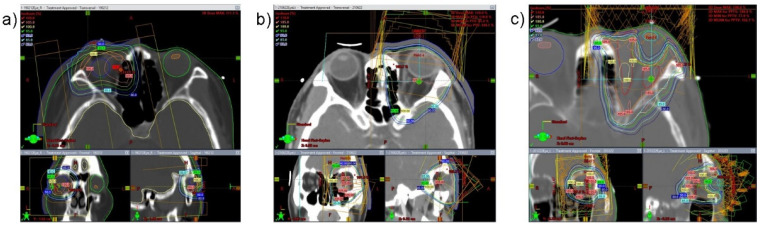
Radiotherapy planning for orbital MALToma: (**a**) electron therapy for conjunctival MALToma with lens shielding and Superflab bolus; (**b**) three-dimensional conformal radiotherapy with non-coplanar beam arrangement for mass invading both the conjunctiva and retrobulbar area; (**c**) intensity-modulated radiotherapy for retrobulbar MALToma.

**Figure 3 cancers-14-04274-f003:**
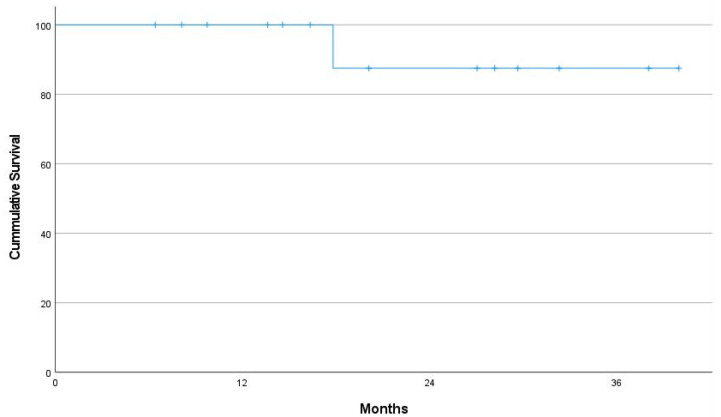
Kaplan–Meier curves for progression-free survival of 14 patients.

**Figure 4 cancers-14-04274-f004:**
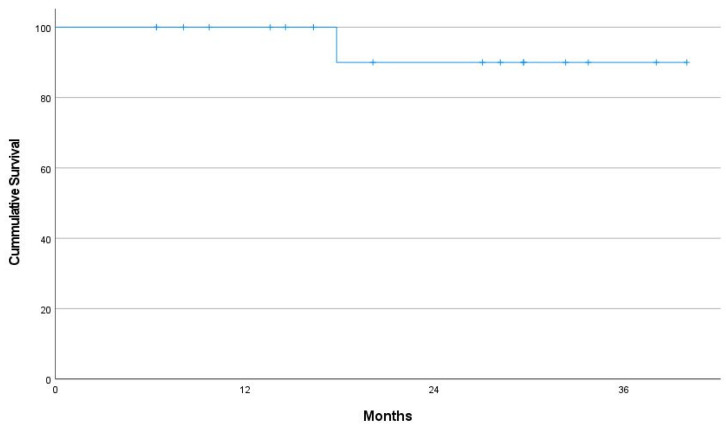
Kaplan–Meier curves for local progression-free survival of 17 eyes.

**Table 1 cancers-14-04274-t001:** Baseline demographic and clinical characteristics of the study patients (*n* = 14).

Characteristic	*n*	%
**Gender**		
Male	7	50.0
Female	7	50.0
**Median Age**	60.5 years	
**Laterality**		
Left	6	42.9
Right	5	35.7
Both	3	21.4

**Table 2 cancers-14-04274-t002:** Baseline characteristics of the enrolled eyes (*n* = 17).

Characteristic	*n*	%
**Laterality**		
Left	9	52.9
Right	8	47.1
**Location**		
Conjunctiva	14	82.3
Retrobulbar	1	5.9
Conjunctiva + retrobulbar	2	11.8
**Visible mass in CT**		
Yes	4	23.5
No	13	76.5
**Hypermetabolic lesion in PET-CT**		
Yes	5	29.4
No	12	70.6
**RT technique**		
2D	14	82.3
3D	2	11.8
IMRT	1	5.9
**Additional RT**		
Yes	5	29.4
No	12	70.6

Abbreviations: RT, radiotherapy; CT, computed tomography; PET-CT, positron emission tomography-computed tomography; 2D, two-dimensional; 3D, three-dimensional; IMRT, intensity-modulated radiotherapy.

## Data Availability

The data presented in this study are available on request from the corresponding author.

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
