# Peer review of "Prospective Study of 4 Gy Radiotherapy for Orbital Mucosa-Associated Lymphoid Tissue Lymphoma (FORMAL)"

_cancers, 2022, doi:10.3390/cancers14174274_

Round 1

Reviewer 1 Report

Although the number of enrolled pts is quite small, the results of the study are really interesting  and deserve future expansion for a possible application in clinical practice. The study is well described. I only suggest to the authors some minor corrections:

-Line 33: the stomach is the most common localization; conjuctiva and ocular adnexa are frequent non-gastric sites and most cases of the disease are stage I

-Line 69: and could receive radiotherapy with curative intent (since this study is not retrospective)

-Line 97: clarify the modality of response evaluation; was it performed by CT and PET scan in case of FDG-avid lesions?

-Line 114: was the complete remission rate after 4Gy the primary end point? Or the complete remission rate at the end of the whole treatment?

-Line 132: please also report the CR and PR percentages

-Line 152: the first sentence of the Discussion is not clear; I suggest a more incisive sentence like this one: the present study showed that 4 Gy ultra-low radiation therapy for orbital MALToma could be safely performed and permits to achieve a complete remission in most patients, reserving a second-planned higher dose only in cases with residual lesions.

-Line 178: In the present study, a complete remission was achieved

-From line 174 to line 183: this paragraph could be moved after line 203

-In the Discussion, I suggest to briefly describe also the safety of the treatment, with limited local side effects

-Line 219: another limit of the study is the short follow up that does not consent to register possible late side effects or relapses

-I suggest to modify the Conclusion according to the above suggestions

Author Response

Dear Editor-in-Chief and reviewers,

I appreciate your attention and helpful comments which contributed significantly to this manuscript. I submit the revised manuscript (cancers-1845472) “Prospective Study of Four Gray Radiotherapy for Orbital Mucosa-Associated Lymphoid Tissue Lymphoma (FORMAL)” to be considered for publication in “Cancers

Please find our responses to the reviewers below.

Reviewer #1: Dear authors,

Although the number of enrolled pts is quite small, the results of the study are really interesting and deserve future expansion for a possible application in clinical practice. The study is well described. I only suggest to the authors some minor corrections:

à We really appreciate your positive feedback.

-Line 33: the stomach is the most common localization; conjuctiva and ocular adnexa are frequent non-gastric sites and most cases of the disease are stage I

à Thank you for your recommendation. We have corrected the manuscript accordingly.

“The most frequently involved non-gastric sites are the conjunctiva and ocular adnexa, and most cases of the disease are stage I”

-Line 69: and could receive radiotherapy with curative intent (since this study is not retrospective)

à Thank you for your recommendation. We have corrected the manuscript as you mentioned.

”and could receive radiotherapy with curative intent.”

-Line 97: clarify the modality of response evaluation; was it performed by CT and PET scan in case of FDG-avid lesions?

àThank you for your helpful comment.

Lugano classification was difficult to apply because many of the orbital MALToma lesions did not have FDG uptake. It seems that there were some errors when writing this manuscript. In most cases, FDG uptake did not exist, so it was evaluated by RECIST criteria in the study design. Compared with the size of the initial mass, changes were measured by gross size for conjunctival MALToma and CT for retrobulbar tumor.

Therefore, we corrected manuscript as you mentioned.

The clinical response was evaluated according to the RECIST 1.1 (Response Evaluation Criteria in Solid Tumours) [11]

-Line 114: was the complete remission rate after 4 Gy the primary end point? Or the complete remission rate at the end of the whole treatment?

à Thank you for your helpful comment.

The primary endpoint was the complete remission rate after 4 Gy radiotherapy.

Therefore, we corrected manuscript as follows:

“The primary endpoint was complete response rate at 6 months after 4 Gy radiotherapy”

-Line 132: please also report the CR and PR percentages

“The complete remission was achieved in 11 (64.7%) lesions, and partial remission was achieved in 6 lesions (35.3%)”

-Line 152: the first sentence of the Discussion is not clear;

I suggest a more incisive sentence like this one: the present study showed that 4 Gy ultra-low radiation therapy for orbital MALToma could be safely performed and permits to achieve a complete remission in most patients, reserving a second-planned higher dose only in cases with residual lesions.

à Thank you for your helpful comment.

We replaced the first sentence with the phrase you suggested.

“The present study showed that 4 Gy ultra-low radiation therapy for orbital MALToma could be safely performed and permits to achieve a complete remission in most patients, reserving a second-planned higher dose only in cases with residual lesions”

-Line 178: In the present study, a complete remission was achieved

àThank you for your recommendation. We corrected the manuscript as recommended.

“In the present study, complete remission was achieved within 6 months at a dose of 4 Gy in 11 of the 17 eyes that were treated”

-From line 174 to line 183: this paragraph could be moved after line 203

àThank you for your recommendation.

I changed the position of the paragraph as you suggested.

-In the Discussion, I suggest to briefly describe also the safety of the treatment, with limited local side effects

àThank you for your recommendation. We have added following paragraph to the Discussion.

In this study, when only a dose of 4 Gy was irradiated, no special side effects were observed during or after treatment because the dose was very low. No serious adverse events of grade 3 or higher were observed. In addition, grade 1 or 2 toxicities were also identified only in patients with residual lesions who received 24 Gy of additional radiation therapy. Kim et al. reported that 24–36 Gy radiation therapy for orbital MALToma might cause dry eyes due to effects on meibomian glands [25]. In addition, no cataracts were observed. In all patients with conjunctival MALToma, lens shielding was used when electron beam therapy was used, and in the case of retrobulbar tumor, the lens was sufficiently protected with IMRT. However, there were cases where both retrobulbar and conjunctiva were involved in two patients. One of these patients achieved complete remission at 4 Gy, and no cataract occurred until 40 months follow-up. Another patient did not develop a cataract until 8 months follow-up after receiving an additional 24 Gy of radiation, which is likely due to the short follow-up effect.

-Line 219: another limit of the study is the short follow up that does not consent to register possible late side effects or relapses

àThank you for your helpful comment. We added the following limitation to the manuscript:

Third, the evaluation of late toxicity was limited because the follow-up of the study was short. Therefore, we plan to conduct additional studies by collecting more patients in the future and extending the follow-up period.

-I suggest to modify the Conclusion according to the above suggestions

àThank you for your recommendation. We added to the manuscript as you suggested.

“Since the number of patients enrolled was too small to obtain the required statistical power and there may be limitations in the evaluation of late toxicity with a short follow-up, we decided to conduct a further investigation by recruiting patients for an additional study period.”

Reviewer 2 Report

In this article authors evaluated ultra-low dose RT (4 Gy) for stage I orbital MALToma. In my opinion the current study is on a topic of relevance and general interest to the readers of the journal.  

This article is well-written but has some critical drawbacks.

First, the biggest problem is the low number of patients with 14 patients enrolled (on planned 28) and 17 eyes. The complete remission rate was >70% but it is not specified how many patients have no residual disease and therefore have not received the additional 24 Gy. I believe that it is difficult to obtain results with good statistical power, but it could be useful to report in a descriptive manner the results of patients who achieved complete response after 4 Gy. 

Considering the number of patients enrolled and the intent of the study (phase II design) in my opinion the results regarding toxicity should have more space in the “results” paragraph. If patients had a specialistic ophthalmological evaluation this should be added to the paper to bring greater value to the results. 

The limits of the study should be explicated in the “discussion” paragraph. 

Pag 4 line 125: “seven eyes were on the left side” do not match with table 2 were left side eyes counted are 9 

Author Response

Dear Editor-in-Chief and reviewers,

I appreciate your attention and helpful comments which contributed significantly to this manuscript. I submit the revised manuscript (cancers-1845472) “Prospective Study of Four Gray Radiotherapy for Orbital Mucosa-Associated Lymphoid Tissue Lymphoma (FORMAL)” to be considered for publication in “Cancers

Please find our responses to the reviewers below.

Reviewer #2:

In this article authors evaluated ultra-low dose RT (4 Gy) for stage I orbital MALToma. In my opinion the current study is on a topic of relevance and general interest to the readers of the journal. 

This article is well-written but has some critical drawbacks.

First, the biggest problem is the low number of patients with 14 patients enrolled (on planned 28) and 17 eyes. The complete remission rate was >70% but it is not specified how many patients have no residual disease and therefore have not received the additional 24 Gy. I believe that it is difficult to obtain results with good statistical power, but it could be useful to report in a descriptive manner the results of patients who achieved complete response after 4 Gy.

à We agree with the reviewer's opinion. We are well aware that the planned number has not been sufficiently reached because we were not able to achieve our expected rate of patient enrollment. Therefore, if the expected results are obtained after analyzing the number of patients enrolled during the originally planned study period, the plan for the future is to enroll more patients and to follow up enough to have statistical power. Nevertheless, we report this study because we believe that reporting a good result as an early result could have clinical implications.

Considering the number of patients enrolled and the intent of the study (phase II design) in my opinion the results regarding toxicity should have more space in the “results” paragraph. If patients had a specialistic ophthalmological evaluation this should be added to the paper to bring greater value to the results.

The limits of the study should be explicated in the “discussion” paragraph.

à

Thank you for your great opinion.

As can be seen from the results, very low numbers of toxicity were reported, which did not require special ophthalmic examination. In addition, cataracts are constantly being examined by ophthalmologists. Most of the patients were irradiated with a dose of 4 Gy, and no side effects were observed, and the cumulative dose for patients who received additional radiation was only 28 Gy, so there were no serious side effects to describe.

Thank you for your recommendation. We added the following to the manuscript as you mentioned in method section:

“Toxicity was assessed based on NCI-CTCAE 4.0.3”

We added the following limitation to the manuscript as recommended.

“In addition, since the toxicity was evaluated with NCI-CTCAE grade, not with an ophthalmic special test, there may be limitations in the interpretation of the results”

Page 4 line 125: “seven eyes were on the left side” do not match with table 2 were left side eyes counted are 9.

à Thank you for your important point. We corrected the manuscript as you recommended.

“Nine eyes were on the left side, and 14 eye lesions involved the conjunctiva only (Ta

Reviewer 3 Report

In their manuscript „Prospective Study of Four Gray Radiotherapy for Orbital Mucosa-Associated Tissue Lymphoma (FORMAL)”, the authors evaluated the effectiveness of very low dose radiotherapy in stage I orbital MALToma. They conclude that ultra-low-dose radiation therapy was safely performed with planned second-line treatment in patients without complete remission. It is an interesting paper and worthy of consideration. I have only a few minor points that should be addressed:

1.       The study group by Laila König et al. have also dealt intensively with the topic. Therefore, the main results and publications of this group should also be cited and debated in the “Discussion”-section (e.g. König et al. Strahlenther Onkol 2016 and 2018 or Trials 2019 – especially under the aspect of an accompanying systemic therapy).

2.       Additional planning figures could describe the radiation therapy more clearly.

3.       The manuscript should be read carefully one more time (e.g. “Ahn arbor”…).

Author Response

Dear Editor-in-Chief and reviewers,

I appreciate your attention and helpful comments which contributed significantly to this manuscript. I submit the revised manuscript (cancers-1845472) “Prospective Study of Four Gray Radiotherapy for Orbital Mucosa-Associated Lymphoid Tissue Lymphoma (FORMAL)” to be considered for publication in “Cancers

Please find our responses to the reviewers below.

Reviewer #3:

In their manuscript „Prospective Study of Four Gray Radiotherapy for Orbital Mucosa-Associated Tissue Lymphoma (FORMAL)”, the authors evaluated the effectiveness of very low dose radiotherapy in stage I orbital MALToma. They conclude that ultra-low-dose radiation therapy was safely performed with planned second-line treatment in patients without complete remission. It is an interesting paper and worthy of consideration. I have only a few minor points that should be addressed:

  1. The study group by Laila König et al. have also dealt intensively with the topic. Therefore, the main results and publications of this group should also be cited and debated in the “Discussion”-section (e.g. König et al. Strahlenther Onkol 2016 and 2018 or Trials 2019 – especially under the aspect of an accompanying systemic therapy).

Thank you for your recommendation. We have added manuscript as you mentioned in discussion section.

. “König et al. [24] pointed out that for the FORT study, it may be difficult to apply to small-sized orbital indolent lymphoma because the size or risk classification of the 2 x 2 Gy treatment group is not appropriate.

König et al. [25] compared 52 patients who received conventional radiotherapy from 24 to 46 Gy with seven patients who received low-dose radiotherapy of 4 Gy. The 2-year local progression-free survival of the patients who underwent low-dose radiotherapy was 100%, and the conventional radiotherapy group was 93.5%, showing good clinical out-comes in both groups.”

  1. Additional planning figures could describe the radiation therapy more clearly.

à Thank you for your recommendation. We have added Figure 2 as you mentioned in method section.

Figure 2. Radiotherapy planning for orbital MALToma, a) electron therapy for conjunctival MALToma with lens shielding and Superflab bolus, b) three-dimensional conformal radiotherapy with non-coplanar beam arrangement for mass invading both the conjunctiva and retrobulbar area, c) intensity-modulated radiotherapy for retrobulbar MALToma.

  1. The manuscript should be read carefully one more time (e.g. “Ahn arbor”…).

We use Editage's English proofreading service and will submit re-edited manuscripts. In the case of “Ahn arbor” - this has been modified:

“if they had a histologically confirmed diagnosis of MALTomas localized in the orbit (Ann arbor stage I)”

Round 2

Reviewer 2 Report

Authors modified the paper according to reviewers request and the article is suitable for pubblicaton